# Different Ventricular Fibrillation Types in Low-Dimensional Latent Spaces

**DOI:** 10.3390/s23052527

**Published:** 2023-02-24

**Authors:** Carlos Paúl Bernal Oñate, Francisco-Manuel Melgarejo Meseguer, Enrique V. Carrera, Juan José Sánchez Muñoz, Arcadi García Alberola, José Luis Rojo Álvarez

**Affiliations:** 1Departamento de Eléctrica, Electrónica y Telecomunicaciones, Universidad de las Fuerzas Armadas—ESPE, Sangolqui 171103, Ecuador; 2Department of Signal Theory and Communications, Telematics and Computing Systems, Universidad Rey Juan Carlos, 28943 Madrid, Spain; 3Arrhythmia Unit, University Hospital Virgen de la Arrixaca, 30120 El Palmar, Spain

**Keywords:** ventricular fibrillation, manifold learning, low-dimensional latent spaces, time-frequency, audio features

## Abstract

The causes of ventricular fibrillation (VF) are not yet elucidated, and it has been proposed that different mechanisms might exist. Moreover, conventional analysis methods do not seem to provide time or frequency domain features that allow for recognition of different VF patterns in electrode-recorded biopotentials. The present work aims to determine whether low-dimensional latent spaces could exhibit discriminative features for different mechanisms or conditions during VF episodes. For this purpose, manifold learning using autoencoder neural networks was analyzed based on surface ECG recordings. The recordings covered the onset of the VF episode as well as the next 6 min, and comprised an experimental database based on an animal model with five situations, including control, drug intervention (amiodarone, diltiazem, and flecainide), and autonomic nervous system blockade. The results show that latent spaces from unsupervised and supervised learning schemes yielded moderate though quite noticeable separability among the different types of VF according to their type or intervention. In particular, unsupervised schemes reached a multi-class classification accuracy of 66%, while supervised schemes improved the separability of the generated latent spaces, providing a classification accuracy of up to 74%. Thus, we conclude that manifold learning schemes can provide a valuable tool for studying different types of VF while working in low-dimensional latent spaces, as the machine-learning generated features exhibit separability among different VF types. This study confirms that latent variables are better VF descriptors than conventional time or domain features, making this technique useful in current VF research on elucidation of the underlying VF mechanisms.

## 1. Introduction

Cardiac fibrillation results in rapid and disorganized contraction of the chambers of the heart. Atrial fibrillation, which occurs at the atria, affects 1% of the population and is the leading cause of cardioembolic cerebrovascular accidents. In contrast, ventricular fibrillation (VF), which occurs at the ventricles, is responsible for more than 80% of sudden death cases in adults [1]. Thus, VF is a life-threatening cardiac arrhythmia in which the coordinated contraction of the ventricular myocardium is replaced by high-frequency disorganized excitation that prevents the heart from pumping blood [2,3,4].

The primary data source used for the study of VF is recorded biopotentials; generally, measurement is performed by surface electrocardiogram (ECG), or occasionally by implantable devices [5,6,7]. Such VF recordings are traditionally described as disorganized biopotentials, and the mechanisms behind this arrhythmia remain unknown. In fact, the causes of the genesis and perpetuation of VF are not easy to determine from either of these biopotential sources. Despite much research, it is unclear to what extent the proposed spectral-based characteristics (such as dominant frequency, unbiased regularity index, or non-polar index) can be sufficiently expressive to identify the existence of different types of VF. Thus, the medical community requires rationale and support for explaining different mechanisms in different patients, allowing clinicians to establish adequate therapy strategies beyond the current state-of-the-art [8].

Machine learning methods have emerged in recent decades, enabling a large number of advances in many fields. In particular, manifold learning techniques deserve special mention within the methods available in current Artificial Intelligence technologies, as they allow visually interpretable representations of the phenomena under study to be obtained [9]. This property can be especially helpful when dealing with problems for which a small set of numerically engineered or neural-network mediated nonlinear features can adequately describe the essence of the input data and show them in a meaningful latent space for the analyzed natural process.

Based on the above, we hypothesized that using manifold learning methods on VF recordings could provide us with a processing tool to distinguish different types of VF. Said tool could be helpful in the current scenarios of VF analysis and recently proposed theories, and could aid in scrutinizing the existence and identification of different mechanisms for this type of arrhythmia. With this objective in mind, we propose the use of autoencoder neural networks (i.e., manifold learners) under unsupervised and supervised learning schemes [10] to provide low-dimensional and interpretable latent spaces in which neural-network mediated features can detect the existence of different types of VF potentials. Our proposed methodology uses a database with experimental recordings from an animal model, in which we certainly know that control cases exist together with different interventions with different drugs to alter the depolarization, repolarization, or tissue conduction properties during VF conditions.

The rest of the manuscript is organized as follows. Section 2 reviews the main concepts related to VF, previous VF research works, and the basic background on autoencoder neural networks. The dataset used in this study is described in Section 3. Section 4 presents the most common time-frequency transforms and audio features. Section 5 discusses the main results obtained using supervised and unsupervised learning autoencoders. Finally, Section 6 concludes this work and envisions relevant future research directions.

## 2. Related Work

VF has been studied for years, and is of great importance as patients can die within a short time due to arrhythmia. Therefore, it is essential to continue advancing the analysis and study of the mechanisms of VF, as well as the possible treatments to combat it, from the use of a defibrillator to less invasive methods such as the use of drugs [11,12]. In addition, machine learning, deep learning, and digital algorithms that analyze other signals (such as voice and audio waveforms) may foster new possibilities in treating this condition. The following subsections aim to summarize these recent findings and technological scenarios.

### 2.1. Current Clinical Research on VF

VF prevents the heart from undergoing orderly and efficient contraction, resulting in the complete loss of cardiac mechanical function. The treatment of this arrhythmia, once initiated and detected, must be managed immediately in order to avoid patient death. The best available therapy is a high-energy shock from an external or patient-implanted defibrillator, whereas to date drug therapy or ablation have not been adopted [13]. High-energy electrical shocks restore the normal rhythm of the heart, ending any chaotic electrical activity and allowing the sinus rhythm to restart and continue. The possibility of restoring the cardiac mechanical function depends on the time elapsed from the onset of the arrhythmia episode. Therefore, nthe implantation of an automatic defibrillator is currently the most successful treatment for patients with uncontrolled VF.

In contrast, pharmacological treatments aim to control fibrillation in a less invasive and more cost-effective way by preventing its occurrence. In principle, drugs would be more desirable if they were as effective as implantable devices. However, antifibrillatory drugs have thus far been unreliable and unsuccessful. Clinical trials to date, in which the mortality rate was higher among treated patients than those receiving a placebo, have highlighted the danger of empirical drug therapies [14]. However, the experimental findings in [15], building on recent advances in understanding cardiac fibrillation, have pointed to a promising new direction for the development of antifibrillatory drugs. These authors demonstrated that VF could be suppressed by pharmacologically altering the electrophysiology of cardiac cells in a specific way that prevents wavebreak instabilities, i.e., by attenuating the restitution curve of the action potential duration. The stabilizing effect of flattening restitution has been predicted based on computer simulations of cardiac action [16].

The VF mechanism and global fibrillatory organization have been recently shown to be determined by gap junction coupling and by fibrosis patterns [17]. In addition, catheter ablation [8,18] has been used in recent years with the aim of treating VF storms, with moderate success. Hence, a need exists to provide more advanced tools to clinical doctors and electrophysiologists in order to help them understand the causes and organization of excitation patterns during VF. Although there is no consensus among experts about a single unifying mechanism for this rapid, uncoordinated, and ineffective heart behavior, evidence exists that, in certain patients, VF activity can appear as organized re-entrant electrical waves sweeping the whole myocardium [19]. These re-entrant wavefronts are often referred to as rotational activity or rotors [20].

### 2.2. Digital Processing and VF Prediction

Digital processing techniques and machine learning technologies have been widely used in studying VF and its characterization. Rotational activity has been used to study the existence of different spectral characteristics in both wavefront behaviors (i.e., rotors and wavebreaks), and the reality of differences between them indicate that they could be due to different mechanisms [19]. The characteristics for studying VF recordings in this research direction have been based on the spectrum, organization indices, and bandwidths of the main spectral peaks [21].

The automatic detection of VF has been widely studied as well, for instance, in [22]. The authors developed an automated diagnostic system for detecting the occurrence of VF in real-time through a time-frequency representation image of the ECG. Their proposed method showed an overall accuracy of 99% using a time-frequency representation based on the Hilbert transform and without any feature extraction stage. On the other hand, Taye et al. [23] used QRS complex shape features instead of the traditional heart rate variability features to predict VF, reaching 98.6% accuracy employing basic machine learning algorithms. Another recent work used the acquisition of cardiac mechanical activity-related signals by non-contact sensors, namely, ballistocardiography, to detect VF [24]. In this last case, a random forest classifier reached 94% accuracy when recognizing the presence of VF.

Other related works have looked for the optimal defibrillation time according to a VF waveform analysis. Thus, Chicote et al. [25] used fuzzy logic and sample entropy to predict defibrillation success and patient survival by minimizing myocardial damage caused by futile defibrillation shocks and minimizing interruptions to cardiopulmonary resuscitation. Similarly, [26,27] employed the amplitude spectrum area and median slope both with and without cardiopulmonary resuscitation for VF waveform analysis prediction of defibrillation shock success. In addition, VF waveform analysis with and without chest compression has been used to predict survival from cardiac arrest [28,29].

Finally, Jeong et al [30] evaluated the optimal length of heart rate variability data, aiming to forecast VF using a simple artificial neural network classifier. About 88% accuracy in terms of heart rate variability was observed in 10- and 20-s time windows. A similar work proposed forecasting ventricular tachycardia one hour before its occurrence using artificial neural networks trained on heart rate variability and respiratory rate variability data [31]. In the same direction, deep learning technologies have recently been used for predicting VF [32] as well as for ECG interpretation [33].

### 2.3. Manifold Learning and Latent Spaces

A latent space representation of structured data is a compressed state that implies a small set of informative and essential variables. This compact representation is a crucial concept in machine and deep learning, as each latent variable learns a fundamental yet hidden feature of the data. In other words, lower-dimensional data models can encode latent variables within a latent space, typically allowing dimensionality reduction and feature extraction to be implemented [10]. Thus, manifold learning is a machine learning scheme based on the assumption that any observed data lie on a low-dimensional manifold embedded in a higher-dimensional space. There are several nonlinear techniques for carrying out such learning, including Hessian Eigenmapping, Local Tangent Space Alignment, Multi-dimensional Scaling, and t-distributed Stochastic Neighbor Embedding [34]. However, autoencoders have emerged as a flexible and valuable manifold learning solution. An autoencoder is an artificial neural network that can be used to learn a lower-dimensional vector representation for higher-dimensional input vector data. In this way, the neural network can be trained to capture essential characteristics of the input data inside each encoding, allowing researchers to subsequently perform quantitative and qualitative analysis on the learned manifolds that represent the intrinsic structure of the input data.

Any autoencoder consists of an encoder, a code, and a decoder, which are related as follows: (i) the encoder compresses the input data $x_{i}$ into an encoding symbol $h_{i}$; (ii) the code $h_{i}$ is the representation of the input data in the latent space or bottleneck of the autoencoder; and (iii) the decoder decompresses the encoding $h_{i}$ and reconstructs the input data as approximated data $\hat{x_{i}}$ [10]. Encoder and decoder usually have symmetric structures, including zero or more hidden layers that are trained in the same way as other artificial neural networks, except that the loss function is used to minimize the difference between the input $x_{i}$ and the output $\hat{x_{i}}$ (or estimation) of the network (see, e.g., [35] for further details).

The typical form of the primary encoder is an affine mapping represented by
(1)$$h_{i}=\phi \left(W_{e}\;x_{i}+b_{e}\right)$$
where $h_{i}\in R^{d}$ is the vector map in the latent space that corresponds to input $x_{i}\in R^{D}$, ϕ(·) is a nonlinear transformation, $W_{e}$ is the weight matrix, and $b_{e}$ is the bias vector. Its corresponding decoder is another affine mapping, provided below:(2)$$\hat{x_{i}}=\varphi \left(W_{d}\;h_{i}+b_{d}\right)$$
where φ(·) is another nonlinear transformation, $W_{d}$ is the weight matrix, and $b_{d}$ is the bias vector. Here, we want to estimate the weights and biases $W_{e}$, $W_{d}$, $b_{e}$, and $b_{d}$ from a set of samples such that $\hat{x_{i}}\approx x_{i}$. Typical loss functions are the mean squared error and the cross-entropy metric.

Although there are various types of autoencoders, we focus here on the simple autoencoder described above with one or more hidden layers. Our main aim is to prove that low-dimensional latent variable embeddings can help to distinguish among different types of VF. In addition, because autoencoders act as unsupervised feature extractors, applying a supervised classifier to the encodings in the latent space is typical, allowing the structure used for classifiers to be simplified and obtaining good accuracy and precision metrics in several applications.

### 2.4. Audio Signals and Psychoacoustic Scales

The analysis of audio signals, especially voice signals, has been developed over the past several decades. An analogy can be made in which the voice and audio production function as the transmitter and the human ear as the receiver. Audio content analysis aims to extract information from audio signals in terms of both the perceptual characteristics of the environment and the person listening to it [36]. Considering the physical nature of audio signals, the best results have typically been obtained by analyzing them in the frequency domain, where the receiver (i.e., the human ear) has a logarithmic response. Thus, researchers have sought to replicate its behavior using critical bands, which can be considered as a filter bank, due to the nature and function of the hearing nerve [36].

A number of so-called scales have been developed using these critical bands and psychoacoustic tests. The most relevant scales in this setting are the Mel, Bark, and ERB (Equivalent Rectangular Bandwidth) scales [37,38], all of which are entirely logarithmic. The number of frequency bands into which each scale can be divided is variable; generally, it is a power of 2. This processing can extract characteristics such as the band or frequency centroids, kurtosis, and more, yielding widespread representations of voice and audio signals. Over the past decades, several of these features obtained with different algorithms have been proven to work and confer advantages in voice and audio processing. More recently, these algorithms and features have been used to characterize, analyze, and model signals from other fields of human knowledge, such as seismic, radar, sonar, and many others [39,40].

## 3. Experimental Dataset and Signal Acquisition

The time evolution of VF latent variables was studied in 27 individuals (anesthetized mongrel dogs) under amiodarone, diltiazem, flecainide, and Autonomic Nervous System Blockade (ANSB). The ECG signals were recorded for 6 min after triggering VF. In Group 1 (five dogs), ECGs were obtained without prior drug administration; Group 2 (five dogs) received amiodarone, 5 mg/kg; Group 3 (seven dogs) received diltiazem, 0.2 mg/kg; Group 4 (five dogs) received flecainide, 2 mg/kg; and Group 5 (four dogs) received propranolol, 0.2 mg/kg and atropine, 0.04 mg/kg. Interested reader may refer to [11] for further details on the data, experimental protocol, and dominant frequency study.

The rationale for choosing these interventions arose from analyzing different molecular and electrophysiological states and alterations during VF episodes, and is explained below for each intervention. *Amiodarone* is indicated for treating recurrent ventricular life-threatening arrhythmias that do not respond to other antiarrhythmic drugs or when other therapeutic alternatives are not tolerated. It effectively converts AF, atrial flutter, or supraventricular tachyarrhythmias to sinus rhythm. This drug works by relaxing the overactive heart muscles, and its main effect on the ionic currents of the cell membrane is a decrease in the exogenous potassium current. Amiodarone reduces the endogenous sodium current and that of the slow L-type calcium channels, and has an inhibitory effect on sympathetic activity. *Diltiazem* is used to treat high blood pressure and control angina (chest pain). Diltiazem belongs to the drug class called calcium channel blockers. It works by relaxing blood vessels, meaning that the heart does not have to pump as hard. In addition, it increases both blood flow to the heart and oxygenation. In addition, it has shown its usefulness in treating associated atherosclerosis and arterial hypertension thanks to its minimal effect on lipid metabolism. *lecainide* is an antiarrhythmic drug used to prevent malignant ventricular arrhythmias and supraventricular tachycardias in patients without structural heart disease. It is a proarrhythmogenic drug, and as such is not considered as a first-line treatment. It slows down the heart’s electrical signals in order to stabilize the cardiac rhythm. Flecainide is a membrane stabilizer that interferes with the fast inflow of sodium during depolarization of the myocardial cells without affecting the duration of the action potential. Finally, the method of *pharmacological blockade of the autonomic nervous system* consists of the intravenous administration of a mixture of propranolol and atropine over the course of three minutes. This protocol has been used for the study and assessment of sinus dysfunction. Its application to healthy subjects has allowed us to appreciate that in the basal state there is a vagal predominance. This predominance of the vagus had been observed in studies in which the administration of atropine resulted in increased heart rate.

Analog-to-digital conversion was performed, and occasional clipping effects were compensated for by a simple signal restoration techniques, mainly based on local median filtering activated when the local standard deviation on 5-ms time windows overpassed a threshold. The resulting sampling rate was 1000 Hz, yielding high-quality VF recordings. Two simultaneous signals were acquired, namely, lead I and lead aVF. We only used the former for the present study, as the latter was notably noisier. Figure 1 shows several examples of recorded signals in different individuals. Panel a shows a representation of 30 s of VF evolution; temporal characteristics such as irregularity and absence of QRS complexes can be observed. Panel b, in logarithmic scale, shows the estimated power spectral density of the periodogram with a rectangular window, showing narrow power peaks around the frequencies from 5 to 10 Hz depending on the signal state at that time instant.

## 4. Proposed Time-Frequency and Audio Features

Neither the time representation nor the spectral domain seems to provide an immediate feature description allowing the control experiments and different interventions to be identified. For this reason, we sought to take advantage of the capacity of machine learning techniques to extract informative features within this context. This section introduces the transformation of VF potentials from time-domain signals into the frequency domain along with a set of valuable features usually employed to characterize audio signals. These signal representations can generate input spaces in VF problems for working with manifold learning, autoencoders, and latent variables.

Previous studies in VF have shown that time-frequency representations often reveal a spectral structure changing with time in VF episodes. Time-frequency signal processing is a set of signal processing methods, techniques, and algorithms in which the two natural variables of time (*t*) and frequency (*f*) are used concurrently. This approach contrasts with other traditional methods in which either time or frequency are used exclusively and independently of the other [41]. Certain signals are better represented by a time-frequency distribution, which is intended to show how the signal energy is distributed over the two-dimensional (t,f) basis. Digital processing of these signals can then exploit the features produced by the concentration of signal energy in these two dimensions instead of only time or frequency. The general form used to represent this description is provided by the equation
(3)$$P_{SP}\left(t,f\right)={\left|\frac{1}{\sqrt{2\pi }}\int_{-\infty }^{\infty }s\left(\tau \right)g\left(\tau -t\right)e^{-j2\pi f\tau }d\tau \right|}^{2}$$
where s(t) denotes the continuous time signal and g(t) denotes a windowing function. For the time-frequency analysis of VF, the type of time-frequency distribution, window type, and window size need to be established, taking into account that due to the Heisenberg uncertainty principle higher frequency resolutions lead to lower time resolution and vice versa.

Hence, the previous consideration can result in a set of features provided by the spectral components estimated for each time instant and its associated window observation according to the current time-frequency representations. On the other hand, additional spectral descriptors are widely used in machine and deep learning applications as well as in perceptual analysis to extract valuable sets of features from the observed time windows of a given signal. Spectral descriptors have been applied to various applications using linear or logarithmic scales such as Mel, Bark, and ERB, which enhance certain frequency bands over others thanks to their filter-bank nature. As VF has often been described in the spectral domain, the spectral descriptors used in audio applications can be helpful in this scenario. Appendix A includes a summary along with the equations for calculating up to eleven features often used in audio processing, which in principle could be used in feature characterization of VF recordings. In audio applications, these eleven spectral characteristics have been proposed to represent several psychophysical properties of speech signals [37]. In our primary case, no filter bank structure and no logarithmic scale are assumed in the spectrum, as VF is often contained in the first 30 Hz of the low-pass band.

## 5. Experiments and Results

### 5.1. Time-Frequency Features

As discussed in previous sections, determining the characteristics of VF in the time domain seems complicated. On the other hand, the most significant amount of VF information is observed from zero to 20 Hz when using frequency-domain representations. We scrutinized a variety of time-frequency distributions, and our preliminary results (not included) showed that the conventional spectrogram with rectangular windowing provides more enhanced details; in particular, the peaks were better determined. In contrast, other windows and time-frequency representations tended to smooth the distributions and create cross-terms. We established a time window size of 2048 samples (about 2 s for signals sampled at 1 kHz). Shorter window intervals were found to lead to spectral estimators with increased variance, while longer window intervals increased bias by mixing the temporal behavior of time instants with different spectral content.

Figure 2 shows the evolution of the VF spectrograms in example subjects for different conditions (windows with an overlap of 50%). This representation shows a patent and smooth-changing structure in the energy distribution over the spectrum; it is notable that about 50 s from the beginning there is a generally pronounced drop in the frequency. Intervention with amiodarone, diltiazem, or ANSB tended to exhibit the shape of a rotated *Y* or fork, although this behavior was barely noticeable for flecainide. The control signal showed a more erratic aspect compared to the others. Furthermore, subjects with flecainide presented a pronounced drop in frequency-band occupation during VF recording, along with a narrower bandwidth, and tended to exhibit more patent energy concentration and increasing spectral band in the first 50 s compared to the other conditions. ANSB had a variable delay at the fork onset and tended to slow VF development compared to the other drugs. With diltiazem, there was a trend towards increasing both the maximum frequency and fork dispersion. In contrast, the frequency range with amiodarone tended to be narrower, and the fork appeared abruptly.

Note that these findings are observable from an adequate representation of a suitable time-frequency distribution. Overall, these recordings and their time-frequency representation reveal the presence of at least two phases in VF development, namely, an initial stage and a later ischemic grade.

### 5.2. Audio-Based Features

We extracted eleven audio features in total using the linear, logarithmic, and mixed frequency scales. Although it may be thought that these types of features might not be relevant, as they are projected in the frequency range of 20 Hz to 20 kHz, [42] previously employed several digital speech and audio processing techniques along with other processing tools and achieved superior results. Considering that most of the information displayed in the spectrum is located in the low frequencies, we restricted the bandwidth to 50 Hz. For example, the time evolution of two audio features is shown in Figure 3 for one example of each different VF condition in our database; Panel a represents the temporal values of the centroid frequency and Panel b represents the spectral dispersion (with an ERB scale and divisions of 256).

These two features can be compared with the results of the time-frequency distributions shown in Figure 2. Because the spectral centroids represent the gravity center of the spectral energy and the spread (sometimes called the instantaneous bandwidth) describes the concentration of the power spectrum around the spectral centroid, a rough degree of information about the spectral shape is provided for narrowband spectra [38]. Both parameters show a change trend after 50 s, as observed from the forked appearance of the time-frequency distributions. In comparing the different subject cases, the value of the spectral dispersion for diltiazem is lower than for flecainide; similar differences are be observed in the time evolution of the other parameters (not shown here for brevity).

### 5.3. Unsupervised Latent Spaces

Unsupervised learning autoencoders were used to scrutinize the structure and properties of the embedding spaces of latent variables generated at the bottleneck of the autoencoder architectures. We hypothesized that, assuming adequacy of the input spaces, short-time windows of VF recordings of different patients with different underlying conditions would exhibit geometrical separation on the latent variable spaces. We additionally hypothesized that this geometrical separation in the latent space could be quantified by a subsequent and independent classification stage in terms of the known types (labels) of each recording.

The space input to the machine learning architectures consisted of different features (time samples, frequency samples, and audio features) obtained from time windows of about 2 s for each VF signal. The whole dataset was split into training and testing groups while avoiding overfitting. The training group consisted of feature vectors from fifteen subjects, with each condition equally distributed. The test set included twelve leftover individuals (one ANSB, three flecainide, two amiodarone, three diltiazem, and three control subjects). Note that while a single individual had all of the time windows either in the training or testing sets, they never had part of their time windows in both groups.

*All conditions and full-time recording*. In the first unsupervised experiment, the VF time signals were used to generate the latent variables. First, the signals were decimated by ten using a low-pass Chebyshev Type I infinite impulse response filter (order 8). Second, the signals were normalized to the maximum of the training set signals and subsequently windowed (204-sample rectangular window with 102-sample overlap). The input space consisted of the time samples in each time window. An autoencoder was generated with four layers (with 16, 3, 16, and 204 neurons in each layer)and trained to predict its input, then the encoder blocks (the two first layers) were extracted. A softmax layer was used to classify the time-window vectors corresponding to each individual in terms of the VF condition used in that experiment. Figure 4a shows the resulting latent space, with a spherical shape and highly mixed conditions, which means that the latent information about different VF types was not retrieved. This result is reinforced by the low classification results obtained with this configuration, shown in Table 1.

In the second unsupervised experiment, the time-frequency distribution of the VF signal yielded the input vectors. First, the VF time signals were preprocessed by mean subtraction and then windowed (2048-sample rectangular windows with 1024-sample overlap). Second, a Welch periodogram was computed for each time window (512-sample rectangular window and 256-sample overlap). Then, the periodogram samples between 0 and 16 Hz were used as input space vectors and normalized to the maximum of the periodogram vectors in the training set. Afterwards, an autoencoder (with six layers of 30, 10, 3, 10, 30, and 67 neurons each) was trained, the encoder segment was extracted, and a softmax layer was again used to quantify the separability in the latent space. Figure 4b shows the resulting latent space, indicating that better separability was achieved by the time-frequency input features as compared to the time-sample input features. Nevertheless, the different drugs noticeably overlapped in the embedding space in pairs, which can be quantified using the classification results in Table 1.

In the third unsupervised experiment, the previously presented audio features were used as the input vector to generate the VF embedding space. First, we computed the eleven audio features for each VF time signal (2048-sample rectangular window with 1024-sample overlap), then each feature was normalized to zero mean and unit variance. Afterwards, an autoencoder (with four layers of 8, 3, 8, and 11 neurons each) was trained. Figure 4c shows that the condition manifolds in the latent space created with audio features presents better separability than with time samples. Although different conditions continue to overlap in the embedding space, as quantified in Table 1, it is relevant that classification accuracy on a five-class classifier often exceeds the accuracy of a random classifier (i.e., 20%).

*Eliminating the ANSB condition and cutting the VF recordings*. The discrimination results in ANSB cases presented limited quality in all three previous schemes. After inspecting the time and spectral representations, and considering the difficulties involved when inducing ANSB in experimental setups, we reconstructed the experiments without including the ANSB individuals. In addition, the time-sample features were discarded for further experiments. After this, the VF spectral and audio features achieved a 2% and 10% overall discrimination enhancement, respectively, as can be seen in Table 1. Although moderate, this increase, as seen in Figure 4d,e, shows embedding spaces with enhanced separability. Hence, ANSB registers seemed to be a non-separable class that mainly introduce noise and distortion to the latent variables. This result indicates that the latent variables method can be sensitive to distortion of recordings in low-dimensional representations of VF. On the other hand, we considered the second half of the signals (the three last minutes), as the effects of these drugs should be better noticed some time after their administration. Increased discrimination capabilities among conditions were again obtained in the VF latent spaces, as shown in Table 1, with the overall accuracy improving by about 19% and 17%, respectively. Figure 4f,g shows that the manifolds for each intervention are more separable than before, meaning that the self-information extracted from the first part of the signals does not show noticeable differences in its dynamics across the different drugs.

### 5.4. Supervised Latent Spaces

The preceding experiments show that discrimination among different types of VF can be achieved in low-dimensional latent spaces generated from autoencoder structures. However, individual accuracy in the classification for each condition can fluctuate depending on the input space (frequency representation or audio-related features). Therefore, we determined whether including a priori knowledge about the segments from each individual could enhance the separability among different VF types in the latent spaces.

We again benchmarked the use of time-domain, frequency-domain, and audio features as input spaces for training the supervised learning autoencoders. In all cases, the VF signals were preprocessed as indicated in the previous subsection. The architectures were as follows: for the time-domain features, we used an autoencoder composed of four layers (16, 3, 10, and 50 neurons each); for the frequency-domain features, the autoencoder had five layers (with 30, 10, 3, 10, and 50 neurons each); and for the audio features, we used four layers (8, 3, 10, and 50 neurons). In all three cases, a classification layer with five or four output neurons was added to the corresponding autoencoder, as mentioned above, in order to achieve multiclass discrimination. Note that the architectures were changed with respect to the previous section, as a set of experiments (not included) showed that using the same architecture for extracting intrinsic features was suboptimal for classification purposes.

Hence, after unsupervised training of the new autoencoders, the weights of the classification layer were initially trained separately, then subsequent fine-tuning of the entire structure (i.e, the autoencoder and the classification layer) was performed to improve the accuracy of the classifier. We call this scheme a supervised learning autoencoder in order to emphasize the classification output layer and the fine-tuning process.

We compared the latent spaces for this architecture before and after applying the fine-tuning training to the whole structure. Figure 5 shows the latent spaces when training with the frequency-domain features before (left) and after (right) the fine-tuning process; it can be seen that the separability among the different types of VF in the latent space is noticeably improved. Table 2 and Table 3 present the accuracy of the unsupervised and fine-tuned trained autoencoders, respectively, for the different input feature configurations. For the unsupervised learning architecture, we confirmed that ANSB had the lowest accuracy and that the overall accuracy improved when it was removed from the training data. In all the cases, flecainide presented the highest accuracy except for the audio features with the second and third datasets. The overall accuracy reached 66.4% using frequency-domain features without ANSB and when only using the second half of the signals. For the supervised learning architecture, ANSB inclusion and removal had a similar effect; Flecainide presented the highest accuracy except for the audio features with the first dataset. The overall classification accuracy reached 74.3% using frequency domain features without ANSB and with only the second half of the signals. Note that fine-tuning increased the average accuracy by almost 9% in the case of frequency-domain features and by more than 10% in the case of audio features.

### 5.5. Training with Complete Sequences

As previously shown, the time-frequency distributions and audio features exhibit differences that can be used for classification tasks. However, psychoacoustic models are relevant elements in speech and audio processing as well. According to Fast et al. [43], the Mel, Bark, and ERB psychoacoustic scales can be used with original audio bands without compromising their behavior. Consequently, these scales and bands were used to compute the previously presented eleven audio features, this time along the whole recordings. The main idea was to classify the types of VF according to the shape presented by the variation of each audio feature throughout the 6 min time frame that these VF episodes last. We suggest that the knowledge of the complete sequence can be more descriptive than the information included in each small sliding time scale window.

Accordingly, each audio feature generated a vector of 1796 values, and the total number of inputs for the autoencoders was 19756. In order to reduce the computational load of this experiment, feature selection based on joint mutual information [44] was applied to reduce redundant or noisy features. Mutual information measures the amount of shared information between each audio feature (input values) and the type of VF (output values); according to this metric, the audio spectral metrics of spread, slope, kurtosis, skewness, and flatness were chosen as the most significant features, as they contained more mutual information than the others.

As in the previous sections, unsupervised and supervised learning autoencoders were trained using the five selected features that produced 8980 input values. The autoencoders used in this subsection only had one input and one latent space layer, where the number of neurons in the latent space varied from 5 to 175. The best results obtained for these variations with the three scales and a number of bands varying from 32 to 256 are presented in Table 4.

Although there were a few sets of examples used for training and testing, because every subject is just an example it is apparent that the classification accuracy varies from 60 to 100% when using the five original classes and the whole VF episodes. In particular, Table 4 presents two results with a 100% success rate. The first one corresponds to the Mel scale with 32 bands and ten neurons using an unsupervised-learning autoencoder. In this case, it can be observed that there was no separation in the latent space and that and the network was overfitted due to the small number of training examples. The supervised version had worse performance. On the other hand, when using a supervised learning autoencoder, a second case with 100% accuracy was obtained with the ERB scale, 256 bands, and 35 neurons. The latent space shown in Figure 6 was analyzed for this result. Whereas high separability was obtained in a moderate set of observations, it it notable that this method provides separation among the ifferent classes.

## 6. Discussion and Conclusions

VF is a complex bioelectric phenomenon that represents a life-threatening arrhythmia. Its mechanism is not fully understood, making it difficult to find new drug and ablation therapies. In fact, recent studies have pointed out the possibility that different mechanisms could exist for VF. Thus, new processing methods should be developed that can help to distinguish biopotentials recorded during different VF episodes. We hypothesized that low-dimensional latent variable spaces could be created in which machine learning-generated features exhibit separability among different VF types, then used several simple manifold learning techniques with this aim. Unsupervised and supervised learning autoencoders found separability among various kinds of VF based on an experimental setup consisting of different drug interventions during induced VF. Supervised learning autoencoders achieved the best accuracy using spectral and audio-based features, although the unsupervised learning autoencoders showed that intrinsically extracted features could be informative enough in these scenarios. In this set of recordings, the proposed method exhibited lower separability during the onset of VF than during the ischemic stage, which is consistent with current clinical knowledge about VF.

The clinical search for VF origin has been addressed recently by the arduous spatial mapping of the heart, which is challenging to perform due to the lethal nature of the disorder and its requiring immediate defibrillation of the patient by electric shocks. To date, one discovery is that electrical tornadoes can form emanating from the arborizing tissue of the ventricular Purkinje cells, which represent only 2% of the total cardiac mass [1]. Thus, the methodology provided in this work can pave the way towards determining whether different types of VF can be present in recordings from patients, using either surface ECG or intracardiac electrograms.

In the past, the organization of biopotentials during VF has been studied using spectral representations, wavelet decomposition, autoregressive models, and metrics such as entropy indices, among many others [45,46,47]. Despite using different features in the time, frequency, or other transformed domains, they were not able to reliably differentiate the different types of VF. However, recent studies have suggested that spectral characterization could yield the best results. While using precise parameters such as fundamental frequency and regularity indices could be informative, doubts have been raised about whether relevant diagnostic information might be discarded in the preprocessing stages [48]. Our method confirms that low-dimensional latent variables are better extracted in spectral domains, whether from the spectrum or from audio-related features.

Here, we have focused on providing signal processing tools for detecting different mechanisms of VF; however, AF mechanisms need to be clarified as well, and they are likely not unique. In both arrhythmias, there is controversy and a need for better tools to scrutinize different mechanisms from patient biopotential signals in order to support clinicians in clarifying and understanding the fibrillatory mechanisms towards the design of new therapies. Although there have been previous works on AF detection using machine learning techniques, they have mainly focused on detecting the presence of AF rhythm without paying attention to interpretability or different types of AF detected from biopotentials. For instance, AF episodes have been detected using heart rate features with support vector machine [49], and AF detection has been addressed with short-term heart rate variability in intelligent wearables and deep convolutional neural networks [50]. These approaches do not follow the clinical evidence that AF may be different in terms of electrophysiological substrates or of the use of different drugs, as shown in a recent prospective study [51] on the different effects of amiodarone vs. metoprolol beta-blockers in heart failure patients with reduced ejection fraction and persistent atrial fibrillation. Differences in clinical responses were found; thus, we expect that biopotentials could be different as well. However, current methods can be limited to this detail in the description. On the other hand, several reviews on VF have compiled the oustanding controversy on different mechanisms for VF or even different phases (initiation, transition, maintenance, and evolution) [52]. Recently, differences have been found in the organization regarding the degree of cardiac fibrosis [17,53]. Although we have not gone into the details of these controversial fields here, our results are consistent with differences in VF biopotential recordings in terms of different conditions (in our case, intervention with different specific drugs). Our results are novel in that, to the best of our current knowledge, no previous works have aimed to provide a tool for detecting different fibrillatory mechanisms in analyzing fibrillatory biopotentials.

We can conclude that different types of VF can be detected in low-dimensional latent variable spaces generated with adequate signal preprocessing on the frequency domain and suitable manifold learning techniques. As far as we are concerned here, the classical approach to VF in data science has typically been episode detection in implantable cardioverter defibrillators. Our approach is focused on more fundamental research that could allow for an understanding of the underlying mechanism behind this life-threatening arrhythmia. According to recent works [54], the most common type of fibrillation, namely, atrial fibrillation, can be created by different mechanisms that in turn lead to distinctive features; in order words, different kinds of AF-originating mechanisms are related to different types of AF. Due to this novelty, we cannot provide differences between the present work and related works; however, these findings could augment our set of potential analytical tools and help pave the way towards better knowledge of the mechanisms of VF as well as to new drug or ablation therapies.

The present work did not cover advanced aspects regarding machine learning and digital processing techniques. We used a straightforward architecture to provide VF latent variables; however, there are many other options. In convolutional autoencoders, the encoder and decoder implement convolution operations with filters and linked weights, thereby preserving relationships among the neighboring data and significantly reducing the number of learnable parameters [55]. Long Short-Term Memory (LSTM) autoencoders use encoding and decoding operations based on recurrent neural networks specifically designed to support sequences of input data [56]. Finally, variational autoencoders generate bottleneck codes describing the average and standard deviation on the latent space for each input vector, thereby yielding compact latent spaces [57]. On the other hand, several time-frequency distributions have been used in the existing literature [41], including the Short-Time Fourier Transform, the Choi–Williams distribution, the Generalized Rectangular distribution, the Margenau–Hill spectrogram distribution, the Minimum Mean Cross-Entropy combination of spectrograms, the spectrogram distribution, and the Smoothed Pseudo-Wigner–Ville distribution. Here, we were only able to apply detailed scrutiny to a smoothed periodogram implementation and provide the input space to the manifold learning procedure. Future work could be devoted to determining whether other manifold learning architectures and other time-frequency distributions can be beneficial to VF latent variable extraction.

## Figures and Tables

**Figure 1 sensors-23-02527-f001:**
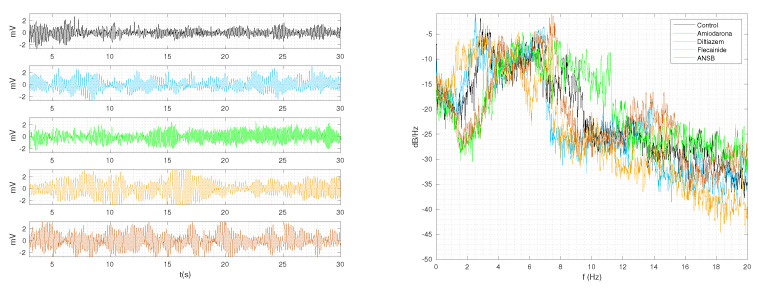
VF in response to various drugs and procedures in the time domain (**left**) and frequency domain (**right**) representations.

**Figure 2 sensors-23-02527-f002:**
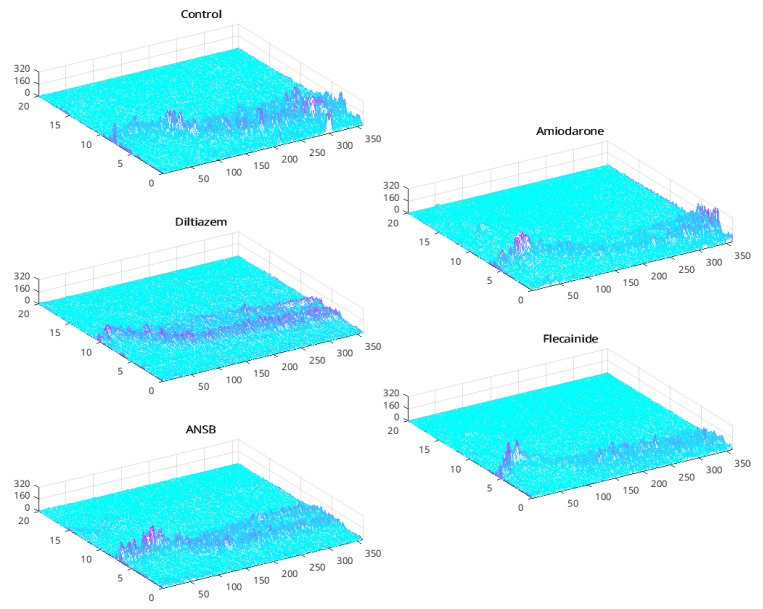
Short-Time Fourier Transform for rectangular windows of size 2048 samples (about 2 s) for several examples of VF signals on the different states and interventions of the experimental conditions.

**Figure 3 sensors-23-02527-f003:**
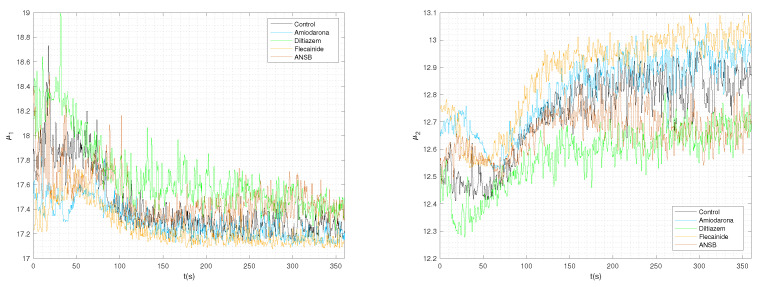
Audio features were obtained for VF recordings in different conditions. The time evolution for each time window is represented by the example of the spectral centroid (**left**) and the spectral spread (**right**) on an example recording of each condition. Despite the variance and fluctuations over time, trends can be observed in different time periods for different features.

**Figure 4 sensors-23-02527-f004:**
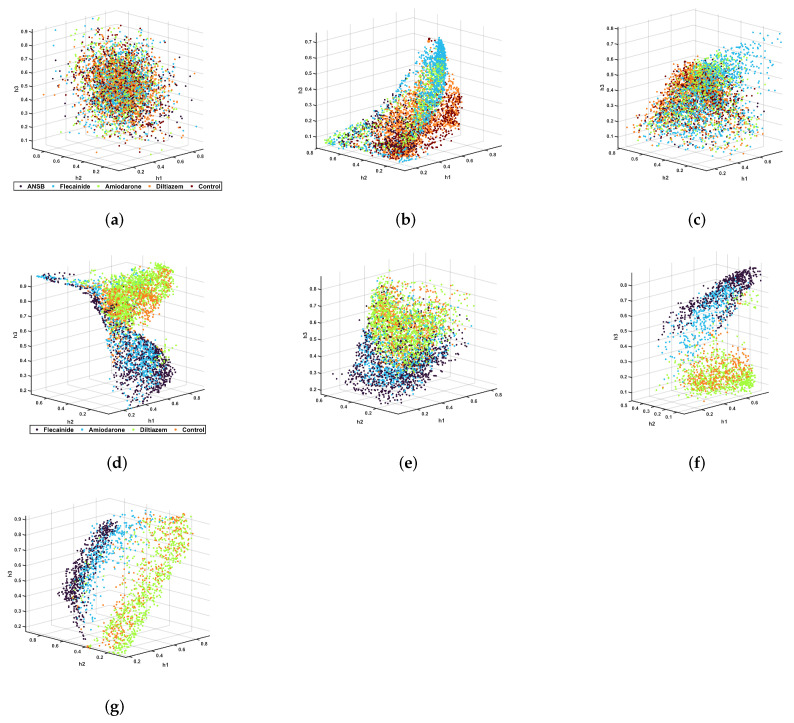
Embedding spaces representing three latent variables for each unsupervised learning scheme. All of the complete (6 min) recordings were used as the windowed-input space in terms of the time (**a**), time-frequency transform (**b**), and audio features (**c**). When the ANSB recordings were not included, this positively affected the structure of the latent variables in the time-frequency transform (**d**) and audio features (**e**). When the ANSB recordings were not included and only the second half of the signals was considered it remained possible to observe different behavior, as seen in time-frequency transform (**f**) and audio-related features (**g**).

**Figure 5 sensors-23-02527-f005:**
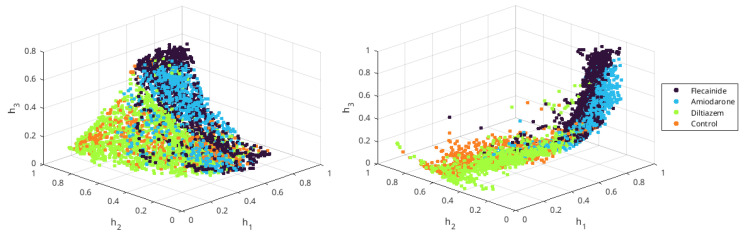
Latent space before (**left**) and after (**right**) fine-tuning using the second dataset and the frequency domain features.

**Figure 6 sensors-23-02527-f006:**
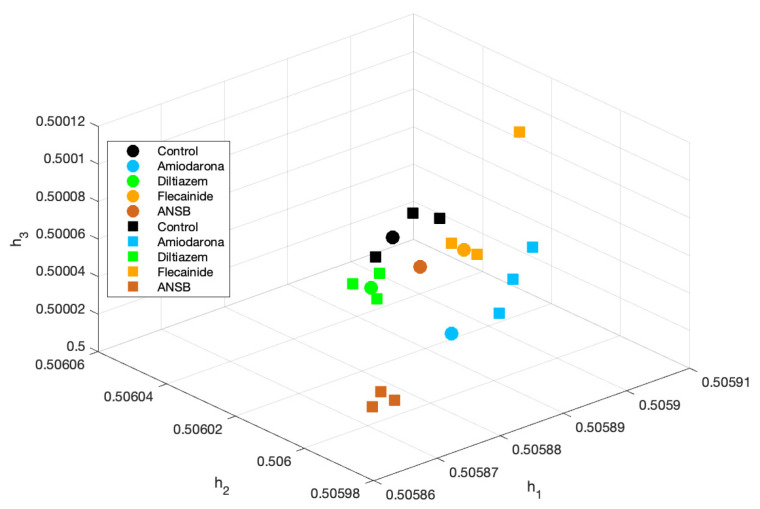
Latent space for the 27 signals using an ERB scale and 256 bands. The squares correspond to the training signals and the circles to the testing signals.

**Table 1 sensors-23-02527-t001:** Accuracy percentage of unsupervised learning autoencoders with several in-order training sets. ANSB is Autonomous Nervous System Blockade, Fle is Flecainide, Ami is Amiodarone, Dil is Diltiazem, and Ctrl is Control.

Data Set	ANSB	Fle	Ami	Dil	Ctrl	Overall
1st set						
Time	100.0	0.0	0.0	0.0	0.0	20.0
Spectral	70.3	59.1	0.0	43.6	0.7	34.7
Audio	64.6	49.7	4.3	29.3	8.9	31.3
2nd set						
Spectra	–	79.1	2.4	61.5	2.7	36.4
Audio	–	37.2	46.0	66.7	15.1	41.3
3rd set						
Spectra	–	96.2	0.0	74.4	51.2	55.4
Audio	–	80.1	42.5	82.0	28.5	58.3

**Table 2 sensors-23-02527-t002:** Accuracy percentage of unsupervised learning autoencoders with several in-order training sets.

Data Set	ANSB	Fle	Ami	Dil	Ctrl	Overall
1st set						
Time	100.0	0.0	0.0	0.0	0.0	8.3
Spectra	16.3	64.1	12.3	48.0	36.9	41.6
Audio	19.1	63.9	10.0	20.6	48.4	34.2
2nd set						
Spectra	–	64.2	16.1	48.3	52.1	47.5
Audio	–	31.0	52.0	23.7	52.7	36.1
3rd set						
Spectra	–	80.3	41.4	71.6	60.3	66.4
Audio	–	17.8	73.9	37.8	61.8	43.3

**Table 3 sensors-23-02527-t003:** Accuracy percentage of supervised learning autoencoders with several in-order training sets.

Data Set	ANSB	Fle	Ami	Dil	Ctrl	Overall
1st set						
Time	100.0	0.0	0.0	0.0	0.0	8.3
Spectra	7.1	60.1	27.4	59.2	45.0	47.4
Audio	23.1	47.0	34.1	70.7	48.4	51.0
2nd set						
Spectra	–	61.7	51.9	60.5	63.0	59.7
Audio	–	73.4	21.4	43.4	44.0	47.7
3rd set						
Spectra	–	83.5	54.6	80.0	69.0	74.3
Audio	–	62.5	25.6	37.9	58.6	46.1

**Table 4 sensors-23-02527-t004:** Best accuracy results for different scales and number of bands.

Scale and Bands	Number of Neurons	Unsupervised Accuracy %	Supervised Accuracy %
Linear	20	60	80
Mel-32	10	100	60
BARK-128	50	60	60
ERB-256	35	60	100

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
