# Peer review of "Different Ventricular Fibrillation Types in Low-Dimensional Latent Spaces"

_sensors, 2023, doi:10.3390/s23052527_

Round 1

Reviewer 1 Report

This article illustrates that different types of Ventricular Fibrillation (VF) can be detected in low-dimensional latent variable spaces, generated with adequate signal preprocessing on the frequency domain and suitable manifold learning techniques. These findings could provide a potential analytical tool that contributes to paving the way towards knowing better the mechanisms of VF and proposing new drug or ablation therapies. There are still some points that need to be added or modified, as follows.

1. In the “related work” section there seems to be a lack of directly relevant paper presentations, i.e., work that examines different types of AF, such as the recently cited literature of the database used, e.g., Ref [1].

[1] Chorro F J, Sánchez-Muñoz J J, Sanchis J, et al. Modifications in the evolution of the dominant frequency in ventricular fibrillation induced by amiodarone, diltiazem, and flecainide: An experimental study[J]. Journal of electrocardiology, 1996, 29(4): 319-326.

2. In sections 5.1-5.3, the authors analyze the different classes and the possible mechanism of VF based on the visualization and classification results. In addition to illustrating the discriminative power of the employed features, this part also draws some conclusions about the mechanisms of AF. We would like to know whether these conclusions are consistent with previous studies or whether these data-driven conclusions are novel.

3. In sections 5.3-5.4, a comparison of the classification accuracy with previous methods is lacking.

4. In Figure 1, the relationship between the two pictures is not “up-down”, “left–right” is more appropriate.

Author Response

Thank you for the reviews , "Please see attached."

Reviewer 2 Report

1. The research motivation and contribution of the manuscript need to be further condensed.

2. The title of the second part of the manuscript, which is proposed to be changed to related work. Because the actual content of this part summarized these recent findings and technological scenarios.

3. The discussion and conclusions are divided into two parts. In addition, the concluding part of the paper is too short.

4. The abstract of the manuscript needs to be readjusted to highlight the innovative points of the paper.

Author Response

(The authors gave the same response as above.)

Round 2

Reviewer 1 Report

I am satisfied how you fulfilled previous requirements. I am suggesting manuscript acceptance.

Reviewer 2 Report

  • I have no other questions.